# Efficient and practical approximation algorithms for advertising in content feeds

## Abstract

Content feeds provided by platforms such as X (formerly Twitter) and TikTok are consumed by users on a daily basis. In this paper, we revisit the native advertising problem in content feeds, initiated by Ieong et al. Given a sequence of organic items (e.g., videos or posts) relevant to a user's interests or information search, the goal is to design an algorithm that maximizes the reward (e.g., clicks) by placing advertisements interleaved with the organic content under two considerations: (1) an advertisement can only be inserted after a relevant content item; (2) the users' attention decays after consuming content or advertisements. These considerations provide a natural model for capturing both the advertisement effectiveness and the user experience. In this paper, we design fast and practical 2-approximation greedy algorithms for the associated optimization problem, in contrast to the best-known practical algorithm that only achieves an approximation factor of 4. Our algorithms exploit a counter-intuitive structure about the problem, that is, while top items are seemingly more important due to the decaying attention of the user, taking good care of the bottom items is key for obtaining improved approximation guarantees. We then provide the first comprehensive empirical evaluation on the studied problem, showing the strong empirical performance of our algorithms.

## Keywords

Newsfeed Advertising, Ad Allocation, Approximation Algorithms, Matching, Externalities

**ACM Reference Format:**
Anonymous Author(s). 2018. Efficient and practical approximation algorithms for advertising in content feeds. In *Proceedings of Make sure to enter the correct conference title from your rights confirmation emai (WWW '25).* ACM, New York, NY, USA, 12 pages. https://doi.org/XXXXXXX.XXXXXXX

## 1 Introduction

A significant share of the current web traffic originates from user-generated content platforms, such as X (formerly Twitter), Facebook, and TikTok [2]. These platforms primarily engage users through their *content feeds*, which display a continuous stream of organic content items, such as social updates or videos, arranged in a carefully crafted order and formatted for infinite scrolling [22]. The main monetization strategy of major social-media platforms is to insert sponsored content in between the content items, such as

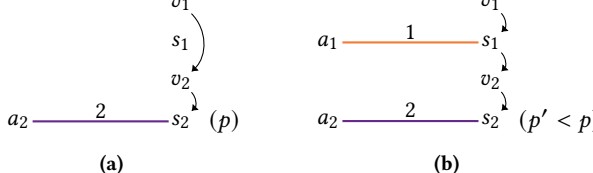

**Figure 1: An illustration of the expected reward being non-monotone with respect to the ad placement. Here $a, v$ and $s$ denote ads, videos, and slots respectively. In the first scenario (a) an ad $a_2$ with reward 2 is allocated to slot $s_2$ after video $v_2$, and a user sees the ad $a_2$ with probability $p$. In the second scenario (b) an additional ad $a_1$ with reward 1 is allocated to slot $s_1$ after video $v_1$. Due to decaying user attention, in (b), the user sees the ad $a_2$ with a probability $p' < p$. Thus, placing an additional ad may lead to a smaller expected reward.**

promoted posts, content seeking higher user engagement, or pay-per-click ads. The sponsored content is often designed to provide a well-integrated look and less intrusive user experience, which is also known as *native* advertising [27]. Advertisers incur a charge every time users interact with sponsored content, and native advertising has evolved into a huge business with a market of about 100 billion USD [19, 23], accounting for nearly two thirds of total display ad spending in the US [8].

The placement of sponsored content within an infinite feed poses a unique allocation challenge as it requires balancing two factors: (a) prioritizing advertisements at the top of the feed, since users will eventually stop scrolling further their feed; and (b) ensuring contextual coherence [29], to boost interaction rates. For instance, an airline advertisement is more attractive when displayed after a travel-related post rather than after a political one. This setting is significantly different from traditional online advertising [7, 20], e.g., search advertising, where ads are sold through auctions for each opportunity, and showing the winning ad is assumed to have no influence on future revenue. In contrast, for native advertising in content feeds, showing an ad reduces the number of items a user will explore. Therefore, if no suitable advertisement fits a specific content, the optimal approach would be to forgo immediate revenue in favor of potential earnings later over the user session. For an illustration, consider Example 1 and Fig. 1.

**Example 1.** *As illustrated in Fig. 1, assume that there is a slot to which an ad can be allocated to, after every organic video. Consider two videos $v_1, v_2$ that are presented to a user in order. Suppose that an ad $a_2$ has been allocated to the slot after $v_2$. The crucial observation here is that placing a new ad $a_1$ before $v_2$ may lead to a loss in the total expected reward over the user session, as it reduces the probability that a user interacts with ad $a_2$.*

Ieong et al. [13] initiated a mathematical formulation for native advertising in content feeds, denoted as the STRMADS problem, where in addition to given rewards for every feasible ad-item pair (e.g., collected through an ad auction), users have decaying attention [6], and may quit browsing with a fixed probability after observing an item or an ad. Under such a model, the STRMADS problem is to maximize the expected total reward over a user session, by suitably deciding a strategy to display ads. Ieong et al. [13] show that there exists a PTAS (i.e., an algorithm that returns nearly optimal solutions) for the STRMADS problem. However, such an algorithm relies on solving expensive combinatorial problems, making it impractical. To the best of our knowledge, the state-of-the-art practical algorithm only achieves a 4-approximation guarantee, that solves the problem by finding a suitable maximum weighted matching (MWM) with cardinality constraints [13].

In this paper, we develop practical and efficient 2-approximation greedy algorithms for the STRMADS problem. To deal with decaying attention, our algorithms exploit a counter-intuitive structure of the problem, namely, while top items are seemingly more important due to the decaying attention, finding a good position for the bottom items is key to obtaining improved approximation guarantees. In addition, to carefully account for the challenging constraints of STRMADS, which requires to allocate rewarding ads while considering the decaying attention of a user, we devise a novel charging scheme based on a non-trivial decomposition of STRMADS's objective function. This result is then used to identify high-quality ad allocation strategies, and leveraged in our proofs to obtain good approximation guarantees.

In addition, to the best of our knowledge, we provide the first comprehensive empirical study on the STRMADS problem. In which we verify the strong empirical performance of our novel algorithms. More specifically, our contributions are as follows.

- We provide an exact greedy algorithm for a special case of the STRMADS problem, where each ad can be displayed more than once.
- We provide two 2-approximation greedy algorithms for the STRMADS problem. The first algorithm uses a greedy criterion guided by the exact marginal gain in revenue, and the second one leverages a lower bound of the marginal gain. The second one is also particularly efficient.
- We provide the first comprehensive empirical study on the STRMADS problem, showing the high-quality ad allocations computed by our novel algorithms.

The rest of the paper is organized as follows. We formally define the problem in Section 2. We characterize the structure of the problem in Section 3. We describe our novel algorithms and prove their approximation guarantees in Section 4. Related work is discussed in Section 5 and extensive experiments are in Section 6. We conclude in Section 7. All the missing proofs are reported in Appendix A.

## 2 Problem definition

In this section, we first present the necessary preliminaries, and then formally define the problems that are studied in this paper.

*Preliminaries.* A graph is *bipartite* if its vertices can be partitioned into two disjoint parts, and edges connect only vertices from different parts. Given an undirected graph, a *matching* is a set of edges so that each vertex appears in at most one edge of the set. For a weighted graph, a *maximum-weight matching* (MWM) is a matching in which the sum of its edge weights is maximized.

A set function $f : 2^E \to \mathbb{R}$ assigns a value to every subset of a given set $E$. A set function $f$ is called *monotonically non-decreasing* if $f(C) \leq f(D)$, for all $C \subseteq D \subseteq E$. Additionally, $f$ is called *submodular* if $f(C + e) - f(C) \geq f(D + e) - f(D)$, for all $C \subseteq D \subseteq E$ and element $e \in E$. Throughout this paper, we use the shorthands $C + e$ for $C \cup \{e\}$ and $C - e$ for $C \setminus \{e\}$.

An algorithm ALG is an *$\alpha$-approximation algorithm* for a maximization problem, if for any instance I of the problem, the solution ALG(I) returned by the algorithm has an objective value that is no smaller than $1/\alpha$ times the value of the optimal solution, denoted with OPT(I) [26]. That is, let $f$ be the objective function of the problem, then it holds that $\alpha f(\text{ALG(I)}) \geq f(\text{OPT(I)})$, for all problem instances I. A *polynomial-time approximation scheme* (PTAS) is an $(1 + \varepsilon)$-approximation algorithm, for any given $\varepsilon > 0$, with running time polynomial in the input size, but possibly exponential in $1/\varepsilon$.

*Problem definition.* We are given a sequence of $m$ items (e.g., videos), and we assume that there is one available *slot* for an ad placement after each item. Suppose also that we are given $n$ ads $A$. To improve the efficacy of the ads, an ad $a_i$ can only be placed after a subset of relevant items $S_i \subseteq V$. A reward $r_{ij} \geq 0$ is then obtained if ad $a_i$ is shown to the user after the $j$-th item, with $j \in S_i$. Throughout the paper, we fix $i$ (resp. $j$) to be the index of an ad (resp. a slot).

To model the decaying attention of the user, our model considers that a user decides to quit browsing (i.e., terminates its session) with probability $q$ after observing every item or ad. Our goal is to decide the allocation of ads to the available slots to maximize the expected reward over the specified model. We use the terms reward and revenue interchangeably. For brevity, we may drop the adjective "expected" if it is clear from the context. More formally, the ad-placement problem is defined as follows.

PROBLEM 1 (STRMADS-R). *We are given a sequence of $m$ items $V$ with one available slot after each item, a set of $n$ ads $A = \{a_i\}$ with associated slots $\{S_i\}$, rewards $\{r_{ij}\}$ for $j \in S_i$, and a quitting probability $q \in [0, 1)$. The goal is to find a mapping $M \subseteq E := \bigcup_{i \in [n]} (\{i\} \times S_i)$ such that every slot can admit at most one ad, i.e., $|\{i : (i, j) \in M\}| \leq 1$ for all $j$, and $M$ maximizes the expected reward*

$$f(M) := \sum_{e=(i,j) \in M} r_e (1 - q)^{j + z(j)}, \tag{1}$$

*where $z(j)$ is the number of slots before slot $j$ containing an ad, i.e., $z(j) = |\{j' < j : (i, j') \in M \text{ for some } i\}|$.*

The STRMADS-R problem explicitly disallows consecutive ads, which helps to avoid ad fatigue and viewer zapping [24]. It is also a common practice to design the ad-allocation strategy as a post-processing operation [17, 28], so that the ranking of content items in input to STRMADS can benefit from state-of-the-art recommenders.

The STRMADS-R problem allows an ad to be displayed multiple times. However, there are scenarios where displaying an ad multiple times is undesirable. To prevent such over-exposure of ads, it is possible to preprocess the slots $S_i$ of each ad $a_i$ and set a limit on the number of slots $|S_i|$. However, such an approach is limited and not always feasible. To provide a rigorous model for such cases, we introduce the following problem variant.

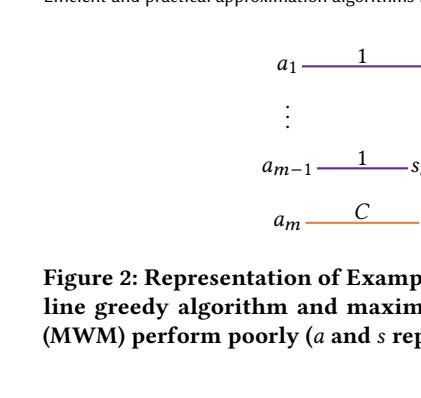

**Figure 2: Representation of Example 2, where a natural online greedy algorithm and maximum weighted matching (MWM) perform poorly ($a$ and $s$ represent ads and slots).**

PROBLEM 2 (STRMADS). *Given the same input as in the STRMADS-R problem, find a* matching $M \subseteq E := \bigcup_{i \in [n]} (\{i\} \times S_i)$ *that maximizes the expected reward $f(M)$ from Eq. (1).*

Note that the STRMADS problem is significantly more general than the previous STRMADS-R problem, as an ad can be displayed multiple times also in STRMADS, by simply generating multiple copies of such an ad. Besides, the STRMADS problem also generalizes the classic maximum-weight matching problem (MWM), obtained from STRMADS by setting the value $q = 0$.

Finally, in Section 6.2 we also discuss how to adapt an algorithm for STRMADS to enforce a limit on the total number of ads to be displayed, which can be useful, for example, to avoid ad fatigue.

## 3 Problem structure and failed attempts

The STRMADS problem was introduced by Ieong et al. [13], who also devised a PTAS algorithm. However, their PTAS relies on exhaustive enumeration of sub-sequences of slots, and flow computations, which is impractical. In this section, we study the structural properties of the STRMADS problem aiming to design a practical algorithm with provable quality guarantees.

Our first step is to view the STRMADS problem as a task of optimizing a specific set function over a bipartite matching. However, as shown in Proposition 1, this specific set function is neither monotone nor submodular. Therefore, the problem cannot be approximated by existing methods for submodular maximization [4].

We then present an example showing that two simple and intuitive heuristics may perform arbitrarily bad. The first heuristic is a standard greedy strategy that prioritizes placing ads in the *top slots*, i.e., the slots appearing at the beginning of the content feed. The second heuristic is to address the problem leveraging the maximum-weight matching (MWM) method. The failure of such approaches, and the problem instance that causes the two heuristics to perform badly inspire the design of our novel algorithms. In the next section (Section 4) we propose a novel backwards greedy strategy that carefully accounts for the placement of ads in *bottom slots*, i.e., the slots appearing at the *end* of the content feed.

**Proposition 1.** *The expected-reward function $f : 2^E \to \mathbb{R}$ in Eq. (1) for the STRMADS problem is neither monotone nor submodular.*

See proof in Appendix A.

Due to the exponentially-decaying attention in the model, a reasonable strategy is to prioritize the top slots. Thus, a logical choice is to employ a greedy algorithm that processes slots in a sequentially increasing order and repeatedly matches the ad with the highest reward to the processed slot. However, as we show below, such a greedy algorithm has an unbounded approximation ratio, even for the easier STRMADS-R problem.

**Example 2** (Being myopic in top slots). *See Fig. 2 for an illustration. For each slot $j = 1, \ldots, m - 1$, we create a dedicated ad $a_j$ with reward 1. For the final slot $j = m$, we create an ad $a_j$ with a large reward $C$. The greedy algorithm assigns each ad in its corresponding slot, and it results in a total expected reward of*

$$\sum_{j=1}^{m-1} (1-q)^{2j-1} + (1-q)^{2m-1}C$$

$$\approx (1-q)/(1-(1-q)^2) + (1-q)^{2m-1}C,$$

*while assigning only the last ad gives a reward of $(1 - q)^{m-1}C$. According to the value of the parameters, the approximation ratio can be arbitrarily bad. For example, with $q = 1/2$ and $C = 2^{2m-1}$, the approximation ratio is about $2^m/2$.*

The instance in Example 2 is also hard for another intuitive algorithm based on maximum-weight matching (MWM). This algorithm finds a MWM for the bipartite graph between ads and slots with appropriately-defined edge weights. That is, every edge $(i, j)$ connecting ad $a_i$ and slot $j$ has a position-biased weight of $r_{ij}(1-q)^j$. Unfortunately, the MWM algorithm fails to capture the decaying-attention effect of the model. It is easy to see that, on the instance from Example 2, the MWM algorithm selects all available edges, like the aforementioned greedy algorithm.

By a careful examination of the bad instance in Example 2, it is clear that to obtain solutions with high expected reward, we cannot only focus on the top slots, or ignore the decaying-attention effect of the model. However, it appears difficult to take care of both ends of the slot sequence. We show in the next section, that both issues can be handled properly by first considering bottom slots, through our novel algorithms.

## 4 Algorithms

In this section, we introduce a novel backwards-greedy algorithm (Algorithm 1, denoted as G-BWD) that carefully handles the bottom slots for ad placement. The backwards-greedy approach addresses the decaying-attention in the model, by iteratively considering sub-problems over suffixes (of the form $j, \ldots, m$, for *decreasing $j$*) of the slots. We show in Theorem 2 that the backwards-greedy algorithm, perhaps surprisingly, finds an optimal solution for the STRMADS-R problem.

On the other hand, it is not straightforward to analyze the G-BWD algorithm for the more challenging STRMADS problem due to the interplay between the decaying-attention effect and the additional matching constraint. To address this issue, we prove a novel decomposition of the expected reward over a matching, which we use to obtain a *non-oblivious* backwards-greedy 2-approximation algorithm (G-BPX in Algorithm 2) for the STRMADS problem, running much faster than G-BWD. More specifically, the G-BPX algorithm adopts a greedy criterion that deviates from the standard marginal-gain greedy criterion (with respect to the underlying objective value). Finally, by leveraging the structural lemmas for the G-BPX algorithm, we provide an analysis for the G-BWD algorithm. We

---

**Algorithm 1:** Backwards greedy (G-BWD)

1   $M \leftarrow \emptyset$; $A_j \leftarrow \{i : j \in S_i\}$ for all $j \in [m]$;
2   **for** *slot* $j = m, \ldots, 1$ *(in a reverse order)* **do**
3     **for** $i \in A_j$ **do**
4       $M_i \leftarrow M + (i, j)$;
5       **if** $M_i$ *is not a valid matching for STRMADS* **then**
6         $M_i \leftarrow (M \setminus \{(i, j') : j' \in [m]\}) + (i, j)$ ;
7       $g_i \leftarrow f_{j-1}(M_i)/(1-q) - f_j(M)$ ;
8     $i^* \leftarrow \arg\max_{i \in A_j} \{g_i\}$;
9     **if** $g_{i^*} > 0$ **then** $M \leftarrow M_{i^*}$;
10   **return** $M$;

---

conclude by also presenting other practical algorithms that can be used to solve STRMADS.

Before presenting our novel algorithms, we introduce a sub-problem of STRMADS, which we refer to as STRMADS-$j$, for a fixed integer $j \in [m]$. In the STRMADS-$j$ sub-problem, the first $j$ items and slots are not considered, i.e., we only consider slots $j+1, \ldots, m$. The resulting objective function for STRMADS-$j$ is,

$$f_j(M) := \sum_{e=(i,j') \in M : j' > j} r_e (1-q)^{j'-j+z_j(j')}, \qquad (2)$$

where $z_j(j')$ is the number of slots after slot $j$ and before slot $j'$ containing an ad, i.e., $z_j(j') = |\{j < k < j' : (i, k) \in M \text{ for some } i\}|$. In particular, $f_0 = f$, while $f_m(\cdot) = 0$.

## 4.1 Solving STRMADS-R optimally

Our backwards-greedy algorithm (G-BWD) for both the STRMADS-R and STRMADS problems is illustrated in Algorithm 1. The G-BWD algorithm returns an optimal solution for the STRMADS-R problem, as we prove in Theorem 2.

The G-BWD algorithm processes the slots in a reverse order, starting from the final slot. At each slot, G-BWD tries to (re-)assign an ad by finding the ad that maximizes the *marginal gain* for the revenue (defined in Line 7). The algorithm performs a (re-)assignment if it results in a positive marginal gain (i.e., increasing the objective function). A matching (or a mapping for STRMADS-R) is then returned after processing all slots.

**Theorem 2.** *Algorithm 1 solves the STRMADS-R problem optimally.*

See proof in Appendix A.

The time complexity for the G-BWD algorithm is $O(|E|)$ for STRMADS-R, and $O(|E|\beta) = O(|E|\min\{m, n\})$ for STRMADS, where $\beta = O(|M|)$ is the time used to compute $f_j(M)$ for $j \in [m]$.

## 4.2 Non-oblivious greedy for STRMADS

The STRMADS problem is more challenging due to the matching constraint. A first idea to address such a problem would be to leverage the G-BWD algorithm, and decompose the revenue of a matching into a sum of marginal gains, one term for each slot. Then, to provide approximation guarantees, we need to connect such marginal revenues to those of an optimal solution for STRMADS. However, such analysis quickly becomes challenging, as a single

---

**Algorithm 2:** Non-oblivious backwards greedy (G-BPX)

1   $M \leftarrow \emptyset$; $\tau_i \leftarrow 0$ for all $i$; $A_j \leftarrow \{i : j \in S_i\}$ for all $j$;
2   **for** *slot* $j = m, \ldots, 1$ *(in a reverse order)* **do**
3     **if** *it exists* $j'$ *s.t.* $(i, j') \in M$ **then** $\sigma(i) \leftarrow j'$;
4     **else** $\sigma(i) \leftarrow j$;
5     $i^* \leftarrow \arg\max_{i \in A_j} \{r_{ij} - \tau_i(1-q)^{\sigma(i)-j}\}$;
6     $g_{LB} \leftarrow r_{i^*j} - q f_j(M) - \tau_{i^*}(1-q)^{\sigma(i^*)-j}$;
7     **if** $g_{LB} > 0$ **then**
8       $M \leftarrow (M \setminus \{(i^*, j') : j' \in [m]\}) + (i^*, j)$; ▷ (re-)assign $a_{i^*}$
9       $\tau_{i^*} \leftarrow r_{i^*j} - q f_j(M)$;
10       **if** $a_{i^*}$ *is re-assigned* **then**
11         $\tau_i \leftarrow r_{ij'} - q f_{j'}(M)$ for every $(i, j') \in M$;

12   **return** $M$;

---

re-assignment (in Line 6) may affect the marginal gain over multiple slots due to the decaying-attention effect.

To avoid such issues, we relate the total revenue to a lower bound of the marginal gains in the above decomposition, that we use to develop a novel greedy algorithm. This results in a 2-approximation non-oblivious backwards-greedy algorithm (G-BPX in Algorithm 2) for the STRMADS problem, note that this approximation ratio is tight for any greedy algorithm. The G-BPX algorithm is called "*non-oblivious*" [16] since it does not select the next ad with respect to the objective function $f$ of Eq. (1).

*The G-BPX algorithm.* The G-BPX algorithm is introduced in Algorithm 2. Similar to the G-BWD algorithm, it processes the slots in a reverse order, starting from the final slot. The key difference is that, at every slot $j = m, \ldots, 1$, it seeks to (re-)assign an ad that maximizes a *lower bound* of the marginal reward, which is

$$\arg\max_{i \in A_j} \left\{ r_{ij} - q f_j(M) - \tau_i(1-q)^{\sigma(i)-j} \right\}, \qquad (3)$$

where $\tau_i$ is defined below, $A_j = \{i : j \in S_i\}$, and $\sigma(i) = j$ if $a_i$ is new to the matching $M$, otherwise $\sigma(i)$ corresponds to the slot previously selected for $a_i$. We prove shortly (in Lemma 4) that Eq. (3) is a lower bound to the marginal reward obtained by assigning an ad at slot $j$.

The term $\tau_i$ represents an *estimate* of the total prior reward provided by ad $a_i$. At the beginning, $\tau_i$ is initialized to be 0 for all $i$. Every time an ad $a_i$ is (re-)assigned to the $j$-th slot, we update its value according to the following rule:

$$\tau_i = r_{ij} - q f_j(M). \qquad (4)$$

It is easy to see that, the first time an ad $a_i$ is assigned to the slot $j$, $\tau_i$ represents its actual marginal reward gain. However, afterwards, if the ad $a_i$ is re-assigned to a different slot $j' < j$, $\tau_i$ deviates from its marginal gain as it does not consider the variation over $f_{j'}(M)$, caused by the withdrawal of $a_i$ from slot $j$. During the execution of G-BPX, it is important to maintain each $\tau_i, i \in [n]$ up-to-update when re-assignments occur. We write $\tau_j = r_{e_j} - q f_j(M)$ when it is more convenient to use the *slot* index $j$, where $e_j$ denotes an edge that is assigned to the slot $j$.

The G-BPX algorithm preforms an ad (re-)assignment if it results in a positive lower bound as from Eq. (3). A matching $M$ is

returned after processing all slots. We show in Theorem 5 that it holds $2f(M) \geq f(M^*)$, where $M^*$ is the matching achieving the optimal solution for $f$, i.e., G-BPX is a 2-approximation algorithm.

*Decomposition.* We now introduce the novel decomposition of the reward of a matching $M$. Let $R_j := f_j(M)$ be the reward of a solution $M$ for the STRMADS-$j$ sub-problem (from Eq. (2)). We have

$$R_j = (1 - q)\left(R_{j+1} + \mathbb{1}[e_{j+1} \in M](r_{e_{j+1}} - qR_{j+1})\right) \quad (5)$$

$$= \sum_{j'=j+1}^{m} (1 - q)^{j'-j}\mathbb{1}[e_{j'} \in M](r_{e_{j'}} - qR_{j'}) \quad (6)$$

$$= \sum_{e=(i,j')\in M:j'>j} (1 - q)^{j'-j}(r_e - qR_{j'}),$$

where $\mathbb{1}[e_j \in M]$ is a $0$–$1$ indicator function taking value 1 if the edge $e_j$, incident to slot $j$, is in the matching $M$. The first equality expresses $R_j$ as a sum of $R_{j+1}$, and the marginal gain obtained by allocating slot $j+1$ with edge $e_{j+1}$. The second equality recursively expands the term $R_{j+1}$, while groups the other terms into a summation. The last equality follows a simple double-counting argument. In summary, $R_j$ is a cumulative sum of marginal gains, computed backwards, of edges in $M$, when there are *no* re-assignments. Notice the similarity between the components in the decomposition in Eq. (6) and the values $\tau_i$ in Eq. (4) (recall that $R_j = f_j(M)$).

We next characterize the behaviour of $R_j$ when a re-assignments occurs in the backwards-greedy algorithm, and connect such results to the greedy criterion in Eq. (3).

**Lemma 3.** *During the execution of the main loop of Algorithm 2, for any fixed $j \in [m]$, the value $R_j$ is non-increasing since the completion of the sub-problem STRMADS-$j$.*

See proof in Appendix A.

*Approximation guarantees.* Next, we explain the novel lower bound presented in Eq. (3). When re-assigning an ad, the exact marginal gain in reward heavily depends on the allocation of all other slots already allocated, due to the decaying attention, making the analysis particularly challenging. Therefore, instead of considering the actual marginal reward, G-BPX seeks a greedy choice that maximizes the non-oblivious lower bound, which simplifies our analysis. We first prove that Eq. (3) (evaluated by G-BPX in Line 6) is a lower bound to the actual marginal reward, provided that every $\tau_i$ in Eq. (4) is maintained up-to-update.

**Lemma 4.** *Denote by $g$ the marginal gain in reward of re-assigning ad $a_i$ from slot $\tilde{j}$ to slot $j$ with $\tilde{j} > j$. Then,*

$$g \geq r_{ij} - qR_j - \tau_i(1 - q)^{\tilde{j}-j}.$$

Finally, we are ready to show the approximation ratio for Algorithm 2.

**Theorem 5.** *Algorithm 2 returns a 2-approximation for the STRMADS problem.*

However, the 2-approximation is tight for both Algorithm 1 and Algorithm 2, and this barrier exists also for the special case where $q = 0$, that is, a MWM instance.

**Proposition 6.** *Algorithm 1 and Algorithm 2 cannot do better than 2-approximation.*

See proof in Appendix A.

The time complexity for the G-BPX algorithm is $O(|E| + m|M|)$ where $|M| = \min\{m, n\}$. The second term is due to the fact that we may need to compute $f_j(M), j \in [m]$ if a re-assignment occurs.

## 4.3 Natural greedy for STRMADS

Algorithm 2 uses a non-oblivious greedy criterion, inspired by the novel decomposition in Eq. (6). We now prove that Algorithm 1 guided by the *exact* marginal reward of an ad is also a 2-approximation algorithm. This seemingly complicated case is a direct consequence of our proof for Algorithm 2.

**Corollary 7.** *Algorithm 1 returns a 2-approximation for the STRMADS problem.*

See proof in Appendix A.

## 4.4 Other practical algorithms

In this section, we introduce various algorithms for the STRMADS problem, including enhanced variants of existing algorithms (from [13]), and multiple practical heuristics. We list all algorithms below, and discuss their important design choices.

*Flow- and matching-based algorithms.* Ieong et al. [13] devised a 4-approximation algorithm FLOW by finding a maximum weighted matching with fixed weights. That is, the matching only considers the decaying effects from items but not ads. The key idea is to reduce the dynamic decaying effect of ad placement by limiting the number of allocated ads (i.e., the matching size) via an additional cardinality constraint. The FLOW algorithm is implemented by a minimum-cost flow, as depicted in its original paper.

We enhance the FLOW algorithm with greedy assignments over the slots not matched by the flow-based procedure, such an algorithm is denoted by FLOWG. We also introduce a natural heuristic MWM, mentioned in Section 3. MWM does not enforce a cardinality constraint to the matching size, and is implemented via a standard maximum-weighted matching algorithm.

*Global greedy algorithm.* We introduce another natural algorithm G-GLB that repeatedly allocates an ad to a slot that maximizes the marginal reward over all allocations, provided the reward being positive. This requires computing the marginal reward of every candidate allocation, with time complexity $O(|E|^2|M|)$, which is expensive. We improve such computation by noting that the marginal reward of any possible allocation is non-increasing over time. This can be used to perform *lazy evaluation* of the marginal reward, i.e., maintaining upper bounds to the actual rewards. That is, we sort all candidate allocations by their rewards in a decreasing order using a heap, and we complete a greedy step if the reward of the top allocation is greater than the upper bounds of all other candidate allocations. Typically, only a few edges need updating every greedy iteration.

*Online greedy algorithm.* In Section 3, we mention an online algorithm G-FWD that allocates an ad in real-time as a user browses its session. Such an algorithm greedily assigns the most rewarding ad to the slot being processed.

Ieong et al. [13] also introduce an online algorithm, which we denote as G-ONL. The idea is to pre-determine a threshold $C_{\text{thr}}$, and for each slot, allocate the most rewarding ad if its reward is greater

than $C_{\text{thr}}$. In our experiments, we test some heuristics to determine the value of $C_{\text{thr}}$.

## 5 Related work

*Native streaming advertising.* The study of sequential ad allocations originates from simple cascade models [1, 15], for which a dynamic-programming algorithm was developed. However, when the reward of an ad depends on the slot position, more sophisticated algorithms are needed [13]. After the work by Ieong et al. [13], several approaches have been proposed, we discuss below.

Gamzu and Koutsopoulos [10] study a variant of native stream advertising, taking into account the distance between consecutive ads to avoid ad fatigue. Yan et al. [28] present a practical solution with an industrial application, by maximizing the revenue while requiring that the total user engagement from organic items exceeds a given threshold. Liao et al. [18] adopt a RL-based model to combine a list of content items and a list of ads to produce a user feed. However, none of these works consider dynamic decay in attention caused by ads.

*Positive externalities in advertising.* On a high level, the StrmAds problem is based on a form of negative externalities, that is, the presence of an ad has a negative effect on future ads. There has been extensive research on the opposite, i.e., positive externalities, in advertising. One notable example is word-of-mouth marketing [11, 14], where it is beneficial to offer products, even for free, to a small group of influencers at the beginning of an ad campaign, to attract more customers.

*Online matching.* There is a rich body of work if externalities are not considered. For example, a standard model of position auctions such as [25] is based on the separability assumption, i.e., the probability an ad receives a click if placed in a position is simply the product of the quality scores associated to the ad and the position, independent therefore of other ads. Under such assumptions, the allocation problem can be treated as a matching problem, for which various algorithms have been developed. We refer the readers to some excellent surveys about matching for more details [7, 12, 21]. Our greedy algorithms are partly inspired by a related streaming algorithm [9]; however, as already mentioned, more sophisticated techniques are needed to handle externalities.

## 6 Experimental evaluation

We provide the first comprehensive empirical study on the StrmAds problem. We do not consider the StrmAds-R problem, as it is a special case of the StrmAds problem, and significantly less challenging given that it can be solved optimally by our G-bwd algorithm.

Our evaluation investigates the following key questions.

(1) How do the algorithms perform by fixing the bipartite graph structure, and varying the weights of the rewards? (Section 6.1)

(2) What is the impact of the problem parameters, such the quitting probability $q$, the number of ads $n$, and slots $m$? (Section 6.2)

(3) How do the algorithms perform for the task of native advertising in content feeds in two realistic scenarios? (Section 6.3)

Our source code is made public for reproducibility and can be found in an anonymous repository.[1]

---

[1]https://anonymous.4open.science/r/StreamAds-code-7351/README.md

We now describe the datasets, baselines, and runtime environment of the experimental evaluation. Note, that all reported results are the average taken over three independent runs.

*Datasets.* To the best of our knowledge, high-quality public real datasets for native advertising are scarce, and existing work mostly uses proprietary data [5, 18, 28]. Hence, we explored two distinct types of datasets for our evaluation. The first type considers random weighted bipartite graphs. Such data is very general, and provides a comprehensive benchmark for the various algorithms considered. The second type of data is obtained by simulating a scenario of native advertising based on real anonymized ad data; more details are in Section 6.3.

*Algorithms evaluated.* We evaluate the performance of our algorithms: the proposed greedy algorithms G-bwd (Algorithm 1) and G-bpx (Algorithm 2), and the practical global greedy algorithm G-glb. Other baselines consist of: two online greedy algorithms G-fwd and G-onl, the flow-based algorithm Flow and its augmented variant FlowG, and the matching-based algorithm MWM. We set the threshold of G-onl to be the best reward at the first slot. We refer the reader to Section 4.4 for a detailed description of the above baselines.

*Environment.* All algorithms are implemented in Python. We adopt a solver for maximum flow and maximum matching from the NetworkX library. All algorithms are executed on a docker image of Ubuntu 22.04. The server is hosted on a Linux system with 48 CPUs of Intel(R) Xeon(R) Gold 6336Y CPU @ 2.40 GHz, 125 GB RAM.

### 6.1 Experiments on synthesized bipartite graphs

In this setting, we first generate a fixed complete bipartite graph over $n = 100$ ads, and $m = 1000$ slots. We evaluate the various algorithms when the input instance has the following three different weighting schemes for the rewards over the edges of the graph: 1) symmetric random weighting, 2) asymmetric random weighting, and 3) finely targeted weighting. Each setting is described in detail below. We also fix $q = 0.1$.

*Symmetric random weighting.* Each edge of the complete bipartite graph has its weight drawn uniformly at random from 1 to 10.

*Asymmetric random weighting.* The random weighting scheme above has symmetric edge weights for different slot positions, which rarely occurs in practice. We break such symmetry and introduce dependencies with slot positions, by the following two methods.

In the first method, edges connecting a top slot have a larger reward. More specifically, the reward $r_{ij}$ for assigning ad $a_i$ to slot $j$ is $r_{ij} = w \cdot (m - j)/m$, with $w$ a random real number in [1, 10], i.e., $r_{ij}$ likely decreases over slot positions. In the second method, edges connecting a bottom slot have a larger reward, that is, $r_{ij} = w \cdot j/m$.

*Finely targeted weighting.* In practice, an ad may be highly relevant to just a few items. To simulate this scenario, for each ad $a_i$ we select a random slot $j$ and set $r_{ij} = 10$, while setting $r_{ij'} = 1$ for all other slots $j' \neq j$.

*Results.* Results are reported in Fig. 3. We first note that the G-onl algorithm has the worst performance, yielding zero reward on most instances. This is likely caused by the fact that its performances heavily depend on the threshold $C_{\text{thr}}$, a parameter that is often hard

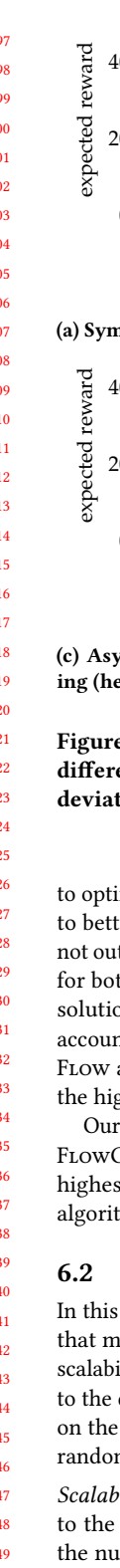

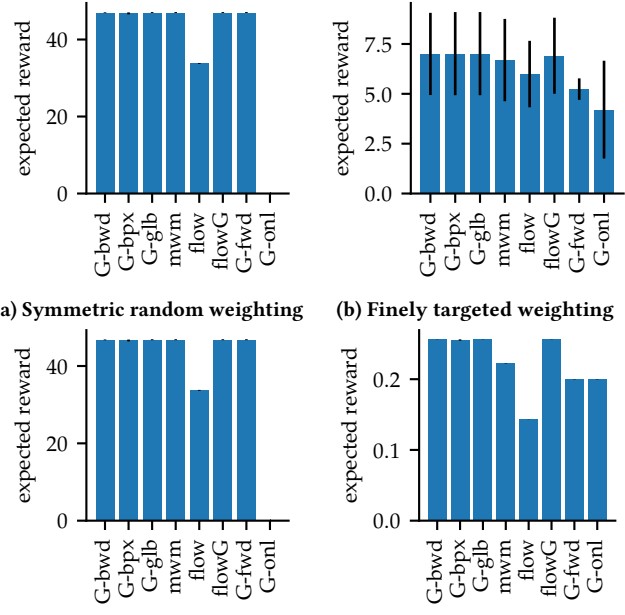

(a) Symmetric random weighting    (b) Finely targeted weighting

(c) Asymmetric random weight-    (d) Asymmetric random weight-
ing (heavy tops)                  ing (heavy bottoms)

Figure 3: Comparisons on synthesized bipartite graphs with different weighting schemes. Error bars indicate the standard deviation.

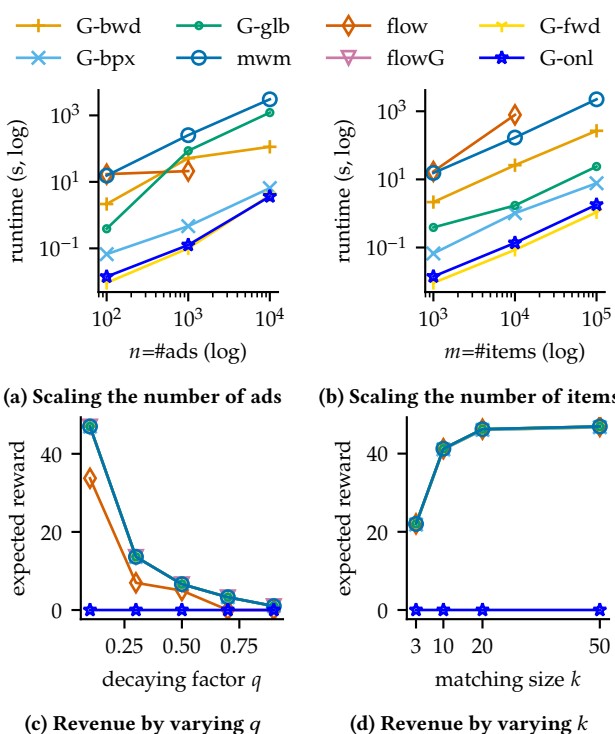

(a) Scaling the number of ads    (b) Scaling the number of items

(c) Revenue by varying $q$    (d) Revenue by varying $k$

Figure 4: Effects of parameters $n, m, q, k$.

to optimize. In the current settings, a lower threshold seems to lead to better solutions. The naïve G-fwd algorithm, as expected, does not output good solutions if there are highly rewarding assignments for bottom slots. In contrast, the MWM algorithm often outputs a solution with reward close to the best observed one, despite not accounting for decaying attention. The 4-approximation algorithm Flow achieves significantly lower expected rewards compared to the highest reward over all algorithms.

Our backwards greedy algorithms G-bwd and G-bpx, G-glb, and FlowG, consistently outperform all other methods and achieve the highest expected reward over all settings, with the global greedy algorithm G-glb providing slightly better solutions.

## 6.2    Ablation study

In this section, we investigate the effect of the various parameters, that may affect the performance of the algorithms. We study the scalability with respect to the size of the bipartite graph, sensitivity to the decaying factor $q$, and to an additional cardinality constraint on the total number of ads to be displayed. We use the symmetric random weighting introduced previously for the edge weights.

*Scalability.* We fixed $q = 0.1$. To test the scalability with respect to the input size, we start with $n = 100$ and $m = 1000$, and vary the number of ads $n$ and the number of videos $m$ separately. The results are shown in Fig. 4a and Fig. 4b, respectively. We set a time limit of one hour for each run. Flow and MWM clearly have the largest running time, as they solve expensive optimization sub-problems. Then, G-glb has also high running time, especially when

$n$, the number of ads, grows, and is less sensitive to the number $m$ of slots due to the lazy evaluation of the rewards, a technique we introduced in Section 4.4. Regarding, G-bpx and G-bwd, while both are backwards-greedy algorithms, G-bpx is much faster than G-bwd, since it uses a lower bound of the true marginal reward, achieving remarkable speedups. The two online algorithms are the fastest, at the expense of significantly lower rewarding solutions.

*Effect of $q$.* We fix the size of the complete bipartite graph, of ads and slots, to be $n = 100$ and $m = 1000$, and we vary the parameter $q$. The result is shown in Fig. 4c. Clearly the expected reward drops as $q$ increases, as users are more likely to quit browsing early in the session. We also note that the Flow algorithm, cannot output a solution when $q > 0.5$; more details are on the original paper [13], making it not practical for general applications.

*Effect of size limit on ads.* Given an integer $k$, we can adapt the algorithms to produce a matching of size *at most* $k$ as follows. We terminate the greedy G-glb and online algorithms after $k$ ad allocations. We set the cardinality constraint of the Flow algorithm to be exactly $k$. While, for all the other algorithms, we iteratively remove one ad at a time whose removal minimizes the loss in the expected reward, if more than $k$ slots are matched in their solution.

We fix $n = 100$, $m = 1000$ and $q = 0.1$. The result are in Fig. 4d. Overall, most algorithms obtain similar performance. Moreover, as $k$ exceeds 20, their revenue reaches a plateau, and further ads bring unnoticeable benefit, in accordance with the value of $q$.

**Table 1: Datasets based on real advertisement. We report:
$n$ number of ads to place, $m$ available slots, $|E|$ number of
edges, the range of the rewards and the value of $q$ used in the
experiments.**

| Dataset | $n$ | $m$ | $|E|$ | $r_e$ ([min - max]) | $q$ |
|---|---|---|---|---|---|
| YouTube | 120 | 14 999 | 1 799 880 | $2.9 \cdot 10^{-5}$ - $3.92 \cdot 10^5$ | 0.1 |
| Criteo | 14 400 | 1 440 | 144 000 | $8.4 \cdot 10^0$ - $1.5 \cdot 10^3$ | 0.1 |

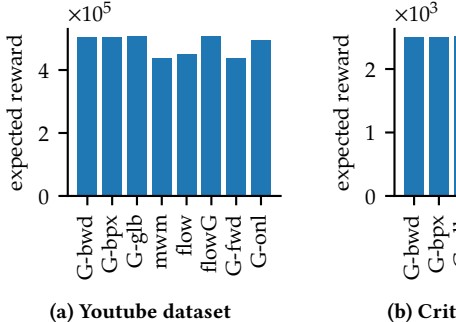

(a) Youtube dataset   (b) Criteo dataset

**Figure 5: Comparisons on simulated native advertising using
real data.**

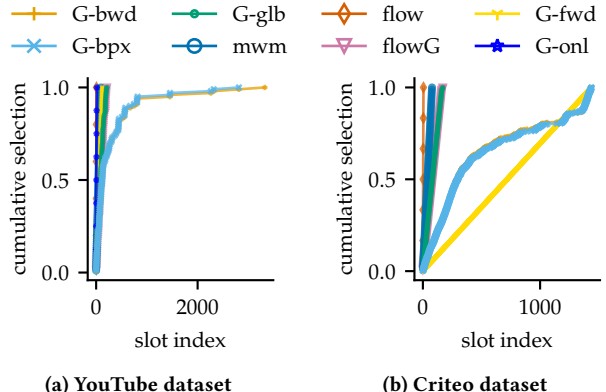

(a) YouTube dataset   (b) Criteo dataset

**Figure 6: Distributions of the selected slot index.**

## 6.3 Simulated native advertising

As mentioned previously, obtaining high-quality advertisement
data is particularly challenging (given its proprietary nature). In
this section we conduct experiments on two datasets built from
real anonymous advertisement data, publicly available.

*Data generation.* Details on how we build instances to our problem
based on two real-world datasets (videos from YouTube[2] and ads
from the Criteo AI Lab[3]) are in Appendix B. Our instances suc-
cessfully preserve the sequential and categorical distribution of
advertisement rewards in the data, when available. A summary of
the key data statistics is reported in Table 1.

[2]https://www.kaggle.com/datasets/sidharth178/youtube-adview-dataset
[3]https://go.criteo.net/criteo-research-kaggle-display-advertising-challenge-
dataset.tar.gz

*Results.* We now discuss the results obtained by the algorithms on
such data. First we report in Fig. 5 the results, in terms of expected
reward for the two datasets. We start by noting that on the YouTube
dataset, the best performing algorithms are G-BWD, G-BPX, G-GLB,
and FLOWG, with G-GLB outperforming all the other algorithms by
a small margin. Surprisingly, the G-ONL algorithm also performs
well. Results for the Criteo dataset confirm a similar trend for the
best performers, but this time together with G-FWD, G-ONL per-
forms poorly compared to others, given by its very sensitive nature.
Such results are in line with what is observed on synthetic data,
confirming the high quality solutions in output to our techniques.

To further investigate the difference in the allocation strategies
produced by the algorithms, we analyzed how the various ads are
placed over the slots. To do this, we report a cumulative distribution
over the slot indices in output to each algorithm, More specifically,
suppose that an algorithm matches $k$ slots with indices $J \subseteq [m]$,
then the cumulative value at index $j$ is $|\{j' \in J : j' \le j\}|/k$. The
results are reported in Fig. 6. On the YouTube dataset, we observe
very different allocation strategies. We first note that methods with
different ad allocation strategies may yield similar expected rewards,
for example G-GLB allocates more slots with larger indices than
MWM despite achieving similar result on the Criteo dataset (see
Fig. 5b). Our backwards greedy methods are the only ones that
allocate ads to slots with large indices. This is due to the backwards
design, which may allocate ads in bottom positions as long as
they are beneficial, even though their utility may diminish later.
In other words, our backwards greedy algorithms achieve a high
recall rate of good allocations. Ads with a diminished reward can be
pruned with little loss, e.g., by the pruning strategy we introduce
in Section 6.2.

As a summary of our experiments, we observe that our proposed
methods report high quality solutions with provable approximation
guarantees (as captured by our analysis) on both synthetic and real
data, and solve the STRMADS problem much more efficiently than
existing techniques.

## 7 Conclusion

In this paper, we provide fast and practical 2-approximation greedy
algorithms for the problem of advertising in content feeds. Our algo-
rithms are faster and theoretically superior than previous methods.
Our analysis relies on a novel charging scheme, derived by care-
fully decomposing and lower bounding the objective function of
the problem. We then provide the first comprehensive empirical
study on the problem, showing the promising performance of our
approaches.

We conclude with a discussion on the limitations of the current
work, and potential future directions. Similar to existing algorithms,
our methods do not work online, which may be limiting for ad
allocation in real-time. Besides, our current work assumes a given
reward for each ad-item pair, and leaves the pricing challenge to
future work. Multiple aspects of the current formulation can be
further refined, for example, a more flexible decaying function, and
more explicit control on the gap between consecutive ads.

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

## A  Missing proofs

**Proposition 1.** *The expected-reward function $f : 2^E \to \mathbb{R}$ in Eq. (1) for the StrmAds problem is neither monotone nor submodular.*

PROOF OF PROPOSITION 1. For simplicity, we consider a special case where, for each ad $a_i$, the rewards $r_{ij}$ are identical, i.e., $r_{ij} = r_i$, for all associated slots $j \in S_i$. We first show that the expected reward is non-monotone. It is easy to see that assigning ads sequentially by the order of the slots increases the expected reward. However, assigning a new ad with a zero reward to an earlier slot decreases the expected reward, as it reduces the probability of subsequent ads of being seen.

We continue to show that the expected-reward function is non-submodular. For any feasible subset $C \subseteq D \subseteq E$, the marginal gain $g((i,j) \mid C) = f(C + (i,j)) - f(C)$ of adding an edge $(i,j)$ into a set of edges $C$ is

$$g((i,j) \mid C) = r_i(1-q)^{j+z(j)} - q \sum_{(i',j') \in C : j' > j} r_{i'}(1-q)^{j'+z(j')}.$$

Compared with $g((i,j) \mid D)$, the first term is clearly non-increasing, but the second term may increase. For example, we have $g((i,j) \mid C) < g((i,j) \mid D)$ by letting $D \setminus C$ be ads with zero rewards placed after slot $j$ *and* before other subsequent items. On the other hand, we also have $g((i,j) \mid C) \geq g((i,j) \mid D)$ when slot $j$ is ranked after every occupied slot in $D$. □

**Theorem 2.** *Algorithm 1 solves the StrmAds-R problem optimally.*

PROOF OF THEOREM 2. The proof is similar to the one by Ieong et al. [13] for finely targeted ads, i.e., $|S_i| = 1$, for all ads $a_i$. The key is to notice that by processing slots backwards, a decision at slot $j$ cannot affect any slot that has not yet been processed, i.e., slots in positions $j' = 1, \ldots, j-1$. That is, the user attention for a slot $j'$ does not depend on ads placed later (in slots $j, \ldots, m$); additionally, every ad can be re-used as there is no matching constraint. Thus, solving optimally the sequence of sub-problems on slots $j, \ldots, m$ with decreasing $j = m, \ldots, 1$, yields an optimal solution to StrmAds-R.

The sub-problem for the final slot (i.e., $j = m$) is trivial, and G-BWD assigns to it the ad with the highest expected reward, if available. Moving backwards to the next slot $j$, G-BWD assigns an ad with the highest reward to the slot $j$ only if it improves the total reward, that clearly results in an optimal assignment for this new sub-problem. The proof immediately follows by the above invariant over the backward processing of the slots. □

**Lemma 3.** *During the execution of the main loop of Algorithm 2, for any fixed $j \in [m]$, the value $R_j$ is non-increasing since the completion of the sub-problem StrmAds-$j$.*

PROOF OF LEMMA 3. At each iteration, $R_j$ remains unchanged if no re-assignment occurs. Hence, consider when an ad $a_i$ is re-assigned from slot $\tilde{j}$ to slot $\tilde{j}'$, and let $\tilde{R}_j$ be the revenue after such a re-assignment. First note that our statement does not regard $R_{\tilde{j}'}$, because the sub-problem StrmAds-$\tilde{j}'$ is completed after the re-assignment. Clearly, $\tilde{R}_j = R_j$ for any $j \geq \tilde{j}$. Now let $j < \tilde{j}$. We prove by induction that $\tilde{R}_j \leq R_j$. Recall that $\tau_j = r_{e_j} - qR_j$, and by design of the G-BPX algorithm it holds $\tau_j > 0$.

First, as a base case, when $j = \tilde{j} - 1$, we have

$$\tilde{R}_j = (1-q)R_{\tilde{j}} \leq (1-q)(R_{\tilde{j}} + \tau_{\tilde{j}}) = R_j.$$

In the inductive step, for $j < \tilde{j} - 1$, we have

$$\tilde{R}_j = (1-q)(\tilde{R}_{j+1} + \mathbb{1}[e_{j+1} \in M - (i,\tilde{j})]\tilde{\tau}_{j+1})$$
$$\leq (1-q)(R_{j+1} + \mathbb{1}[e_{j+1} \in M]\tau_{j+1}) = R_j,$$

where $\tilde{\tau}_j = r_{e_j} - q\tilde{R}_j$. The inequality follows since $\tilde{R}_{j+1} \leq R_{j+1}$ holds regardless of $e_{j+1}$ being in $M$ or not. This completes the proof. □

**Lemma 4.** *Denote by $g$ the marginal gain in reward of re-assigning ad $a_i$ from slot $\tilde{j}$ to slot $j$ with $\tilde{j} > j$. Then,*

$$g \geq r_{ij} - qR_j - \tau_i(1-q)^{\tilde{j}-j}.$$

PROOF OF LEMMA 4. The marginal gain $g$ of re-assigning ad $a_i$ from slot $\tilde{j}$ to slot $j$ is a sum of two terms. The first term is the loss of removing edge $\tilde{e} = (i,\tilde{j})$, and the second term is the marginal reward of adding the new edge $(i,j)$. By Eq. (6), we have that

$$R_j - \tilde{R}_j = \sum_{j'=j+1}^{m} (1-q)^{j'-j}\left(\mathbb{1}[e_{j'} \in M]\tau_{j'} - \mathbb{1}[e_{j'} \in M - \tilde{e}]\tilde{\tau}_{j'}\right)$$

$$= \tau_i(1-q)^{\tilde{j}-j} + \sum_{j'=j+1}^{\tilde{j}-1} (1-q)^{j'-j}\left(\mathbb{1}[e_{j'} \in M](\tau_{j'} - \tilde{\tau}_{j'})\right)$$

$$\leq \tau_i(1-q)^{\tilde{j}-j},$$

where $\tilde{R}_j$ is the reward after the removal, and $\tilde{\tau}_j = r_{e_j} - q\tilde{R}_j$. The last two steps follow from Lemma 3. The claim follows,

$$g = \tilde{R}_j - R_j + r_{ij} - q\tilde{R}_j = (1-q)(\tilde{R}_j - R_j) + r_{ij} - qR_j$$
$$\geq r_{ij} - qR_j - \tau_i(1-q)^{\tilde{j}-j+1} \geq r_{ij} - qR_j - \tau_i(1-q)^{\tilde{j}-j}$$

□

**Theorem 5.** *Algorithm 2 returns a 2-approximation for the StrmAds problem.*

PROOF OF THEOREM 5. We prove the claim by induction on slots $j \in [m]$ following the same backward ordering (i.e., $j = m, \ldots, 1, 0$) adopted by Algorithm 2. Let $\text{ALG}_j$ be the solution of Algorithm 2 before performing the $j$-th iteration (i.e., having only processed the slots in positions $m, \ldots, j+1$)[4], and $\text{OPT}_j$ be the optimal solution to StrmAds (i.e., OPT) ignoring the first $j$ slots. Let their objective values for the sub-problem StrmAds-$j$ be $R_j := f_j(\text{ALG}_j)$ and $R_j^* := f_j(\text{OPT}_j)$, respectively. And also let the marginal revenue in $R$ be $g_j = R_{j-1}/(1-q) - R_j$ at the $j$-th slot, and similarly in $R^*$, $g_j^* = R_{j-1}^*/(1-q) - R_j^*$. We then write $\Gamma_i := \tau_i(1-q)^{\sigma(i)-j}$, for each ad $a_i$ matched in $\text{ALG}_j$.

Let $\tilde{j}$ be smallest $j$ such that it holds $R_{\tilde{j}} \geq R_{\tilde{j}}^*$. Note that $\tilde{j}$ exists, as $R_m = R_m^* = 0$. If $\tilde{j} = 0$, the statement trivially follows. Otherwise, we assume the following hypothesis: for every $j < \tilde{j}$, we can charge marginal revenue $g_j^*$ of $\text{OPT}_j$ to both $g_j$ and $\{\Gamma_i\}$ in $\text{ALG}_j$, while maintaining the invariant that every $\Gamma_i$ (corresponding to ad $a_i$) in $\text{ALG}_j$ is used at most once among all iterations. This immediately implies

$$2f_j(\text{ALG}_j) = \sum_{j'>j} g_{j'}(1-q)^{j'-j} + \sum_{e=(i,j') \in \text{ALG}_j} \Gamma_i$$

---

[4]for $j = m$ there are no such processed slots, while if $j = 0$ then $\text{ALG}_j$ corresponds to the output of Algorithm 2.

$$\geq \sum_{j' > j} g_{j'}^* (1-q)^{j'-j} = f_j(\text{OPT}_j)$$

by the decomposition in Eq. (6).

For $j = \tilde{j}$, since $R_j \geq R_j^*$, it is sufficient to consider only the marginal gains $\{g_j\}$, as it holds $R_j \geq R_j^*$. Now, for the next smaller $j$ in an inductive step, we have the following cases.

**Case 1.** $\text{OPT}_{j-1} = \text{OPT}_j$, that is, OPT does not include any new ad for its $j$-th slot. If our ALG also does not select any item for the $j$-th slot, then the inductive step clearly holds.

Otherwise, notice that Algorithm 2 (re-)assigns an ad only if $g_{LB} > 0$ by Lemma 4. Hence, the overall revenue (i.e., $R_{j-1}/(1-q)$) only increases, and therefore our hypothesis holds also for this case.

**Case 2.** $\text{OPT}_{j-1} = \text{OPT}_j + e^*$, where $e^* = (i^*, j)$, that is the optimal solution assigns ad $i^*$ to the $j$-th slot.

**Case 2.1.** If our ALG (re-)assigns ad $i$ to slot $j$, i.e., matching the edge $e = (i, j)$, then by the greedy criterion (Eq. (3)), we have

$$r_e - \Gamma_i \geq r_{e^*} - \Gamma_{i^*}.$$

Therefore, we can use both $\Gamma_{i^*}$ and $g_j$ to charge for $r_{e^*}$. That is,

$$g_j + \Gamma_{i^*} \geq r_e - \Gamma_i - qR_j + \Gamma_{i^*} \geq r_{e^*} - qR_j^* = g_j^*,$$

where the first inequality follows by Lemma 4, and the second follows by the greedy rule and the fact that $R_j < R_j^*$ (as $j < \tilde{j}$). Note that if ad $a_{i^*}$ was not matched in $\text{ALG}_j$ then $\Gamma_{i^*} = 0$, or otherwise, we increase the number of charges on $\Gamma_{i^*}$ by one.

**Case 2.2.** $\text{ALG}_{j-1} = \text{ALG}_j$. The greedy choice and its inequalities from Case 2.1 still apply, but fail to produce a positive lower bound. That is, $g_{LB} = r_e - \Gamma_i - qR_j \leq 0$ for each $e = (i, j)$. Therefore, it is sufficient to only pay $\Gamma_{i^*}$ for this case.

In Case 2, we use each $\Gamma_i$ at most once because OPT contains at most one edge incident to ad $a_i$, given the matching constraint. Furthermore, $\tau_i$ is non-decreasing after re-assigning either ad $a_i$ (by design of G-BPX), or other ads $a_{i'}$ (by Lemma 3), so the payments in prior iterations remain valid, completing the proof. □

**Proposition 6.** *Algorithm 1 and Algorithm 2 cannot do better than 2-approximation.*

PROOF OF PROPOSITION 6. Fix $q = 0$, and then STRMADS is reduced to a maximum weighted matching problem (MWM). It is well known that a greedy algorithm cannot do better than 2-approximation for MWM. Concretely, let $m = 2$. Create two ads $a_1, a_2$ with slots $S_1 = \{1, 2\}$ and $S_2 = \{2\}$, respectively. Set rewards $r_{11} = r_{22} = 1$ and $r_{12} = 1 + \epsilon$. Thus, a backwards-greedy algorithm yields a revenue of $1 + \epsilon$ by assigning $a_1$ to the 2-nd slot, while the optimum assignment yields 2. The ratio approaches 2 for an arbitrary small $\epsilon$. □

**Corollary 7.** *Algorithm 1 returns a 2-approximation for the STRMADS problem.*

PROOF OF COROLLARY 7. The proof is similar to Theorem 5, except that we need a different inequality for the Case 2 therein. Though Algorithm 1 does not use the values $\tau_i$, we use such values here only for the analysis, and assume that Algorithm 1 updates the values $\tau_i$ as from Theorem 5. Recall that $\Gamma_i := \tau_i (1-q)^{\sigma(i)-j}$.

Suppose that at slot $j$, $\text{OPT}_{j-1} = \text{OPT}_j + e^*$, where $e^* = (i^*, j)$. If our ALG (re-)assigns edge $e = (i, j)$, then by the greedy criterion,

$$g_i \geq g_{i^*}$$

$$r_{ij} - qR_j - \kappa_{ij} \geq r_{i^*j} - qR_j - \kappa_{i^*j},$$

where $g_i$ denotes the marginal reward of (re-)assigning ad $a_i$, and $\kappa_{ij} := r_{ij} - qR_j - g_i$. By Lemma 4, we have for any $i$,

$$g_i \geq r_{ij} - qR_j - \Gamma_i \implies \Gamma_i \geq \kappa_{ij}.$$

Therefore, we can use both $\Gamma_{i^*}$ and $g_j$ to charge for $r_{i^*j}$. That is,

$$\begin{aligned} g_j + \Gamma_{i^*} &= r_{ij} - qR_j - \kappa_{ij} + \Gamma_{i^*} \\ &\geq r_{i^*j} - qR_j - \kappa_{i^*j} + \Gamma_{i^*} \\ &\geq r_{i^*j} - qR_j^* - \kappa_{i^*j} + \Gamma_{i^*} \\ &\geq r_{i^*j} - qR_j^* = g_j^*, \end{aligned}$$

where the inequalities follow by the greedy rule, the fact that $R_j < R_j^*$, and Lemma 4, respectively.

The claim follows by charging every $g_j^*$ to $g_j$ and $\{\Gamma_i\}$, and noting that every $\Gamma_i$ is used at most once among all iterations. We omit the details for the other cases, as they follow from Theorem 5. □

# B Native advertisement data

In this section we describe how we built data used for our experimental evaluation on native advertisement, i.e., the setting in Section 6.3.

*YouTube data.* The YouTube data we considered is formed by a set of videos $\{v_1, \ldots, v_m\}$, characterized by: (1) the video category, i.e., $C(v_i) \in \{C_1, \ldots, C_\ell\}$, where $\ell = 8$; and (2) the number of "ad views" for each video, which we use as a proxy for the reward. To generate the data, we first obtain a random browsing session, i.e., a permutation $v_1', \ldots, v_m'$ of the videos, through the following browsing model. A user starts from a randomly-chosen video $v_1'$. With probability $p = 0.5$, the user selects another randomly chosen video of the same category $C(v_1')$, or otherwise the user randomly selects a previously unseen video from a different category. The process is iterated until a permutation of all videos is obtained.

We assume that there are $r = 15$ advertisers, providing $1, \ldots, \ell$ ads, i.e., one for each category $k \in [\ell]$. We compute the reward $r_{ij}$ for ad $a_i$ after video $v_j$, where $i \in [r\ell]$ and $j \in [m]$, as follows. First, for each different category $C_k$ with $k \in [\ell]$, over all the videos belonging to $C_k$, we compute the average "ad views" $\mu_k$ and its standard deviation $\sigma_k$. We then assume that the rewards are normally distributed, i.e., $r_{ij} \sim \alpha_k |\mathcal{N}(\mu_k, \sigma_k)|$, where $k = C(v_j)$, and parameter $\alpha_k = 0.8$ if the ad and the video share the same category, i.e., $C(a_i) = C(v_j)$, or $\alpha_k = 0.01$ otherwise, which captures a higher reward for ads targeted to related videos. Hence in the final data each ad $a_i$, $i \in [r\ell]$ can be placed after each video $v_j'$, with the reward $r_{ij}$ computed as above.

*Criteo data.* The data consists of a chronologically ordered *sequence* of displayed ads collected over one day. Each of the 48 millions ads recorded has 13 numerical features (capturing the engagement of users with each displayed ad), that we clustered into $k = 100$ categories using the $k$-means algorithm. Besides, a reward can be computed for each ad, as a linear function of its features. We simulate the following browsing session over a full day: a user is browsing a website and an ad can be displayed to its session after one minute of content observed on the website, that is there are exactly $m = 1440$ slots to which ads can be assigned. We then create $b = 144$ blocks of ads (which may correspond to different

advertisers), and for each block, we assume $k$ (non-existential) ads, i.e., one for each cluster. We then associate ads in each block to 10 random slots among $m$. Then, for each block-slot assignment we add connecting edges, that is, suppose the ads in block $h \in [b]$, with indices $a_{(h-1)k+1}, \ldots, a_{hk}$ are associated to slot $j$ then we add edges of the form $(a_{(h-1)k+i}, j)$ for $i \in [k]$. Then, if there exists an edge between $a_i$ with $i \in [bk]$ and slot $j \in [m]$, then the reward $r_{ij}$ is assumed to be the *average reward*[5] of all ads (from the original data) of the same category as $a_i$[6] displayed over the $j$-th minute; otherwise, $r_{ij} = 0$. In this way, we capture the reward distribution over both clusters and time, in real-world data.

---

[5]More formally let $a_i = (a_i^1, \ldots, a_i^{13})$ be ad $a_i$ with its features. Then we compute, for each ad it maximum engagement $\max_{h=1,\ldots,13}\{|a_i^h|\}$, which we further multiply by a factor 10 if the ad was clicked by a user. Such value is then averaged to compute the actual average reward.

[6]among the $k$ categories obtained trough $k$-means.

