# OpenReview forum: "Efficient and practical approximation algorithms for advertising in content feeds"
_ACM.org/TheWebConf/2025/Conference — WWW 2025 Poster_

### Official Review · Reviewer_7oMC · 2024-11-25

**Novelty:** 4
**Technical Quality:** 7

**Review:**

This submission studies the problem of advertising in content feeds (such as X/Twitter and TikTok), formalized as the StrmAds problem (originally posed by Ieong et al.). In this setting, a sequence of content items (e.g. videos or posts) is presented to users in a specific order. Ad slots are available between content items, and rewards are associated with placing ads after relevant content items. (E.g. a travel ad may receive a high reward following travel-related posts.) User attention is assumed to decrease as they progress through the feed. This is modeled by a quitting probability, i.e. after each item/ad, the user will quit browsing with probability q.

The goal of the advertiser in this setting is to maximize the expected reward from all ad placements in a user session. This requires balancing ad effectiveness (i.e. ads placed earlier are more likely to be seen) and user experience (i.e. irrelevant ads could lead to low rewards).

In a simpler version of the problem where an ad can only appear once (called StrmAds-R), the authors provide a backwards greedy algorithm which obtains the optimal solution. The algorithm works by placing slots in reverse order. At each slot, it places the (feasible) ad which maximizes the marginal gain in expected reward.

For the general StrmAds problem, the authors show that the backwards greedy algorithm achieves a 2-approximation, which improves on the previous best result (a 4-approximation). However, its analysis is more challenging in this general setting due to the interplay between the decaying-attention affect and the additional matching constraint present in this version of the problem. To address this issue, the authors prove a decomposition of the expected reward from a matching, and use this to obtain a non-oblivious backwards-greedy algorithm which is also a 2-approximation. The authors also study other practical algorithms for the StrmAds problem, such as flow- and matching-based algorithms, a global greedy algorithm, and an online greedy algorithm.

The experiments focus on three questions: how the algorithms perform across different problem structures/reward schemes, the impact of problem parameters (e.g. quitting probability, number of ads, number of slots) on performance, and algorithm performance on real-world data. The real-world datasets are based on YouTube videos and ads from the Criteo AI Lab.

Strengths:

This paper introduces several new algorithms which improve on the previous state-of-the-art for the StrmAds problem, a practical and impactful problem which studies ad placement in content feeds. The analysis of one of the algoirhtms introduces a novel reward decomposition for the StrmAds problem, which may contribute to the community’s broader understanding of this problem. Moreover, the authors provide an optimal solution for a simpler version of the problem. The algorithms are computationally-efficient, and the experiments they run are comprehensive.

Weaknesses:

The proposed algorithms are designed for the offline version of the problem, in whihc it is assumed that the entire sequence of content and ads are determined up front. In reality, user behavior/rewards are dynamic, and so there is potentially much to be gained from considering an online strategy for assigning ads to slots.

**Questions:**

n/a

**Reviewer Confidence:**

3: The reviewer is confident but not certain that the evaluation is correct

**Scope:**

4: The work is relevant to the Web and to the track, and is of broad interest to the community

---

### Official Review · Reviewer_KaFD · 2024-12-01

**Novelty:** 3
**Technical Quality:** 3

**Review:**

New greedy algorithms for optimizing advertising placement in content feeds are proposed. However, significant modifications are still required in terms of the applicability of the algorithms and the experimental datasets.

**Questions:**

Generality in real - world scenarios: The model assumes that the reward values of advertisements and content are static, which may simplify the complexity of dynamic reward values in actual scenarios. How can the model be extended to support dynamic reward values, such as adjusting dynamically based on user interactions or market changes?

Lack of diversity in datasets: The experimental data mainly focuses on synthetic data and small - scale real data, lacking verification with large - scale and heterogeneous data. Do the authors plan to test the algorithms on larger - scale and more diverse datasets, including proprietary data in the industrial field?

Insufficiency in baseline comparison: What is the basis for the selection of baselines in the experimental part? Although the paper includes multiple baseline methods, there is a lack of comparison with machine learning methods (such as reinforcement learning). Do the authors consider comparing the method with reinforcement learning - based advertising allocation algorithms to verify the performance advantage?

Optimization of algorithm complexity: Although the proposed algorithms have good approximation guarantees in theory, the time complexity of the algorithms may still be a key issue in practical applications.

**Reviewer Confidence:**

4: The reviewer is certain that the evaluation is correct and very familiar with the relevant literature

**Scope:**

3: The work is somewhat relevant to the Web and to the track, and is of narrow interest to a sub-community

---

### Official Review · Reviewer_msts · 2024-12-02

**Novelty:** 6
**Technical Quality:** 6

**Review:**

Summary

This paper studies the following problem faced by advertisers in “content feeds” like Instagram or TikTok, where there is a stream of content where ads can be inserted, but the user has some chance of leaving after each piece of content. There are n slots in a feed where ads can be inserted. There are m different ads. Each ad i can go in some subset S_i of the slots, and has some value to the advertiser if the viewer sees it in a specific slot. The viewer progresses through the slots (and inserted ads) in order, having a fixed probability of leaving after every item they process (similar to the cascade models in bandits). Ads can either be reused or not, depending on the details of the model.

The authors prove the following results:
- First, the simplest greedy algorithm you might try (progess through the slots in order, assigning the best possible ad to each slot) is sub-optimal, possibly by a very large approximation factor (e.g. in cases where there is a very high value ad that can only go near the end).
- In the case where you can reuse ads, there is a simple backwards induction greedy algorithm that runs in polynomial time (process suffixes of the feed in increasing size).
- This backwards induction greedy algorithm can be adjusted to the setting where you can’t reuse ads by reallocating “previously” allocated ads (here “previously” refers to ads later in the feed, but allocated previously in the backwards induction). The authors prove that this algorithm results in a 2-approximation to the optimal allocation, by analyzing a slightly more complicated algorithm that uses a slightly different non-oblivious greedy criterion. They also prove this 2-approximation factor is tight for both algorithms.
- Finally, they experimentally compare these algorithms to other reasonable benchmarks (e.g. the forward greedy algorithm mentioned above, a threshold based algorithm defined in prior work, ...) in some synthetic and semi-synthetic (generated via YouTube data) settings. They show that the algorithms they introduce noticeably outperform all other algorithms in both classes of settings.

Evaluation

I overall enjoyed reading this paper (it was probably the most interesting of all the papers I reviewed for TheWebConference this year). Display advertising in content feeds is a very important and algorithmically non-trivial problem. This paper studies a mathematical model of this problem, and designs some algorithms with non-trivial guarantees (I think it is particularly non-obvious that the backwards induction algorithm gives a 2-approximation guarantee in the matching case, although someone with more expertise in submodular optimization might find it less surprising). Finally, although this model is (by necessity) somewhat stylized, I found it to not be excessively so, and the algorithms the authors propose strike me as the types of algorithms that one could actually implement (variants of) in practice.

Also, although I am generally more of a theorist, I found the experimental results pretty interesting. I was not expecting the backwards induction approach to be meaningfully better than the obvious forward greedy algorithm in practice, if only because my intuition was that in practice only the first couple items should matter (examples like the counter-example should not really happen). But it seems like the backwards induction approaches are noticeably better even on some of the semi-synthetic datasets, which is pretty interesting.

The paper was overall well-written and easy to read.

**Questions:**

Feel free to reply to any aspect of the above review.

**Reviewer Confidence:**

3: The reviewer is confident but not certain that the evaluation is correct

**Scope:**

4: The work is relevant to the Web and to the track, and is of broad interest to the community

---

### Official Review · Reviewer_9yBL · 2024-12-02

**Novelty:** 5
**Technical Quality:** 5

**Review:**

The paper studies the problem of native advertising: how does a social media platform like Twitter or TikTok balance displaying ads with ensuring context coherence? Requiring this tradeoff comes from the following core difference between native advertising and classical online advertising, where ads are shown assuming no impact on the future revenue. In contrast, in native advertising (e.g., on Twitter), showing a completely irrelevant ad might prevent the user from engaging with the content entirely, thereby resulting in a loss of revenue.

The work of Ieong, Mahdian, and Vassilvitskii initiated the mathematical model of this problem, with a reward-maximizing objective that must also account for users' decaying attention. Using combinatorial algorithms, they got a $4$-approximation algorithm for this problem. This paper's contribution is a practical and efficient $2$-approximation algorithm for this problem.

The authors view the problem as a task of optimizing a specific set function over a bipartite matching. However, this function lacks monotonicity and submodularity, thus necessitating novel approaches beyond submodular maximization. Specifically, the authors propose algorithms that also pay attention to items at the bottom of the ad feed, thus taking a non-myopic view to designing the ad feed and making sure that the decaying attention span of the viewer doesn't penalize the overall reward.

**Questions:**

I think this is a nice result and well explained overall. I was wondering if the authors see any connections to the literature on positive linear programs (packing and covering LPs) that could be of use to improve the efficiency of their algorithms.

**Reviewer Confidence:**

3: The reviewer is confident but not certain that the evaluation is correct

**Scope:**

4: The work is relevant to the Web and to the track, and is of broad interest to the community